# Molecular and Cellular Neurobiology of Spreading Depolarization/Depression and Migraine: A Narrative Review

**DOI:** 10.3390/ijms252011163

**Published:** 2024-10-17

**Authors:** Eiji Kitamura, Noboru Imai

**Affiliations:** 1Department of Neurology, Kitasato University School of Medicine, Sagamihara 252-0329, Japan; ekitamura@med.kitasato-u.ac.jp; 2Department of Neurology and Headache Center, Japanese Red Cross Shizuoka Hospital, Shizuoka 420-0853, Japan

**Keywords:** migraine, CSD, CGRP, trigeminal nervous system, intranasal insulin-like growth factor 1, vagus nerve stimulation

## Abstract

Migraine is a prevalent neurological disorder, particularly among individuals aged 20–50 years, with significant social and economic impacts. Despite its high prevalence, the pathogenesis of migraine remains unclear. In this review, we provide a comprehensive overview of cortical spreading depolarization/depression (CSD) and its close association with migraine aura, focusing on its role in understanding migraine pathogenesis and therapeutic interventions. We discuss historical studies that have demonstrated the role of CSD in the visual phenomenon of migraine aura, along with modern imaging techniques confirming its propagation across the occipital cortex. Animal studies are examined to indicate that CSD is not exclusive to migraines; it also occurs in other neurological conditions. At the cellular level, we review how CSD is characterized by ionic changes and excitotoxicity, leading to neuronal and glial responses. We explore how CSD activates the trigeminal nervous system and upregulates the expression of calcitonin gene-related peptides (CGRP), thereby contributing to migraine pain. Factors such as genetics, obesity, and environmental conditions that influence the CSD threshold are discussed, suggesting potential therapeutic targets. Current treatments for migraine, including prophylactic agents and CGRP-targeting drugs, are evaluated in the context of their expected effects on suppressing CSD activity. Additionally, we highlight emerging therapies such as intranasal insulin-like growth factor 1 and vagus nerve stimulation, which have shown promise in reducing CSD susceptibility and frequency. By elucidating the molecular and cellular mechanisms of CSD, this review aims to enhance the understanding of migraine pathogenesis and support the development of targeted therapeutic strategies.

## 1. Introduction

Migraine is one of the most common neurological disorders, affecting over one billion individuals worldwide, with an estimated global prevalence of approximately 15–20% [1,2,3]. It is particularly prevalent among individuals of working age, typically between the ages of 20 and 50 years, and it is the leading cause of disability among women under 50 years of age, significantly impacting their social and professional lives [1,4]. Despite its high prevalence and substantial socioeconomic burden, the pathogenesis of migraine remains incompletely understood [5].

Migraine is a complex neurological condition characterized by recurrent episodes of severe headache, often accompanied by nausea, photophobia, and phonophobia. In approximately one-third of patients, these headaches are preceded by an aura phase lasting between 5 and 60 min [2]. The most common type of aura is visual, experienced by more than 90% of migraineurs with aura [2]. The aura has traditionally been closely associated with cortical spreading depolarization/depression (CSD), a wave of neuronal and glial depolarization that propagates across the cortex [3,6,7,8,9,10,11]. CSD is not only implicated in the aura phase but is also thought to play a critical role in triggering migraine headaches by activating trigeminovascular pathways.

Understanding the role of CSD in migraine pathophysiology is crucial for developing targeted therapeutic interventions. While significant progress has been made in elucidating the mechanisms underlying CSD and its relationship with migraine, gaps remain in our knowledge of the factors that modulate an individual’s susceptibility to CSD and how this can be leveraged for treatment [6,7]. Identifying these factors could lead to personalized medicine approaches, improving prophylactic and acute treatment strategies for migraine sufferers.

### Aim of the Review

This narrative review aims to provide a comprehensive overview of the current understanding of migraine pathophysiology with a focus on the role of CSD. We explore the factors influencing the threshold for CSD initiation, including genetic predispositions, constitutional factors, and environmental triggers. Furthermore, we discuss therapeutic interventions that target CSD, examining both established and emerging treatments. By synthesizing current research on CSD and its impact on migraine, we aim to enhance the understanding of migraine pathophysiology and support the development of more effective, personalized therapeutic interventions.

## 2. History of the Discovery of CSD

The understanding of CSD in relation to migraines has developed over decades, with significant contributions from various researchers. The first breakthrough came in 1941, when Lashley conducted a comprehensive analysis of the visual phenomena associated with migraine. Lashley observed that the visual field disturbances in the occipital lobe’s visual cortex moved at approximately 3 mm/min [12]. This finding marked a pivotal moment in the history of CSD research, as it linked visual aura with brain activity.

A few years later, in 1944, Leão first reported CSD in a rabbit model of epilepsy [13]. He observed that electrical stimulation of the brain surface led to a transient suppression of brain activity (as seen on EEG), which propagated across the cortex at a similar rate to Lashley’s visual field disturbances. This phenomenon was also associated with the expansion of the pial artery, further establishing a link between cortical electrical activity and vascular changes.

In 1958, Milner proposed that CSD could be the underlying cause of migraine aura, highlighting the strikingly similar propagation rates between CSD and the visual aura seen in migraines [14]. This hypothesis set the stage for future research into the connection between CSD and migraines.

Fast-forward to 1981, when Olesen et al. measured regional cerebral blood flow (rCBF) using the xenon-133 intra-arterial injection method in patients experiencing migraine with aura (MWA) [15]. They followed six cases from the normal state into the prodromal phase and, in three cases, further into the headache phase. Their findings revealed that during the prodromal phase, all patients displayed occipitoparietal rCBF reduction (oligemia), which gradually spread anteriorly over 15 to 45 min. In four patients, severe headache was present concomitantly with oligemia and without any signs of hyperemia or uneven brain perfusion. Interestingly, in three patients, the attacks were initiated by focal hyperemia. Additionally, the normal increase in rCBF during cortical activities such as hand movement and speech was impaired in six patients. These observations suggested that the traditional vasospastic model of migraine was too simplistic, providing key evidence linking CSD to both vascular and neural mechanisms in migraine.

In 2001, Hadjikhani et al. used functional magnetic resonance imaging (fMRI) to detect CSD-like signal changes during migraine auras. These signal alterations originated from V3A in the occipital cortex and spread at a rate of 3.5 mm/min, further corroborating the earlier findings [16].

Migraine can be divided into MWA and migraine without aura (MWoA). Initially, CSD was believed to occur only in MWA. However, more recent studies have shown that CSD-like changes in cerebral blood flow may also occur in MWoA. For instance, a study by Woods et al. employed positron emission tomography (PET) imaging to detect bilateral hypoperfusion in the occipital, temporal, and parietal lobes during spontaneous migraines without aura [17]. Similarly, Denuelle et al. found reduced blood flow in the occipital, parietal, and temporal cortices during migraine attacks without aura [18]. Yet, not all studies agree, as some have failed to detect significant changes in cerebral perfusion in MWoA [19,20].

Beyond its role in blood flow changes, CSD has also been linked to neuroinflammation and the activation of matrix metalloproteinase-9 (MMP-9) [21]. Although there is indirect evidence of neuroinflammation in migraines—such as intracranial plasma extravasation and gadolinium enhancement—clear demonstrations of these phenomena in migraine cases remain elusive [21,22]. Recent research has explored the use of superparamagnetic iron oxide nanoparticles (ferumoxytol) to identify the blood-brain barrier (BBB) dysfunction during neuroinflammation [23]. While these nanoparticles can be absorbed by immune cells like microglia, their inability to cross an intact BBB suggests that, in cases of MWoA and MWA, the BBB might remain largely undisturbed [24,25]. However, elevated MMP-9 levels are frequently associated with migraine occurrences, indicating a potential, albeit subtle, role for BBB disruption in migraine pathology [26,27].

In summary, CSD plays a fundamental role in migraine pathogenesis, especially during the aura phase. It is a pervasive and dynamic phenomenon observed across various models, from animals to humans, offering valuable insights into the mechanisms of both aura and headache generation in migraines [9].

## 3. Microglia and Neuroinflammation in CSD

Microglia and neuroinflammation play a pivotal role in the pathogenesis of migraine and are significant contributors to the consequences of CSD [24,25,28,29,30,31]. CSD is characterized by a wave of depolarization followed by the suppression of neuronal activity, which propagates across the cortex. This process can lead to the release of various proinflammatory mediators, including cytokines and reactive oxygen species, which in turn activate microglia [29].

Once activated, microglia contribute to maintaining an inflammatory environment, potentially exacerbating neuronal excitability and pain transmission pathways. This is thought to play a role not only in the migraine aura but also in migraine pain [28,29]. Microglial activation leads to the release of proinflammatory cytokines such as interleukin-1β (IL-1β), tumor necrosis factor-α (TNF-α), interleukin-6 (IL-6), as well as the upregulation of enzymes like inducible nitric oxide synthase (iNOS). These substances further activate surrounding glial cells and neurons, amplifying the inflammatory response [32] (Figure 1).

This creates a feedback loop that sustains and amplifies the inflammatory response, ultimately lowering the threshold for subsequent CSD events. This increases the likelihood of recurrent CSD and potentially contributes to the development of chronic migraine [24,30].

Furthermore, the inflammatory mediators released by microglia can influence neuronal excitability and potentially affect the integrity of the BBB. Although some studies suggest that the BBB remains intact during attacks of migraine without aura, microglial activation and the resulting neuroinflammation may cause subtle changes in the BBB’s permeability that are not easily detected by conventional imaging techniques [24]. Over time, these subtle changes may contribute to the pathophysiology of chronic migraine by facilitating the infiltration of peripheral immune cells into the brain, thereby exacerbating the inflammatory response [30].

The chronic activation of microglia and the ongoing neuroinflammation that they perpetuate are believed to be key factors in the transition from episodic to chronic migraine [28]. The continuous release of inflammatory cytokines can prolong neuronal hyperexcitability and lower the threshold for migraine triggers [30]. Therefore, targeting microglial activation and neuroinflammation may represent a promising therapeutic strategy for managing migraines.

## 4. Animal Studies and CSD

Although CSD plays an important role in migraine development, it is not exclusive to this condition. CSD was originally discovered in the context of epilepsy, and subsequent studies have demonstrated that it can occur in various other neurological disorders, including cerebral infarction, subarachnoid hemorrhage, and traumatic brain injury [8,33,34,35]. This finding is significant because it highlights the generalized nature of CSD as a fundamental neurophysiological response to brain injury and hyperexcitability, rather than being a phenomenon unique to migraines.

The occurrence of CSD in diverse neurological conditions suggests that it may serve as a common pathway of pathological neuronal activity. In cerebral infarction, for example, CSD has been linked to the progression of ischemic damage by exacerbating neuronal depolarization and excitotoxicity. Similarly, in traumatic brain injury, CSD is associated with secondary injury processes that involve neuronal and glial dysfunction. This broader understanding of CSD underscores its potential as a therapeutic target, not only for migraines but also for conditions where excessive brain excitability or a disruption of cellular homeostasis is a contributing factor. Investigating CSD in these various contexts may offer new avenues for therapeutic interventions aimed at stabilizing brain activity and preventing long-term damage across multiple neurological diseases.

In animal models, CSD can be reliably induced using techniques such as topical potassium administration, glutamatergic agonists, microembolization, or physical stimuli (like a pinprick to the brain’s surface using a needle tip) [3,36,37,38,39,40,41,42,43,44].

Figure 2 shows a rat model of CSD induced by the application of KCl drops onto the cortical surface [41,45]. Although this model can induce CSD with relative stability, it is invasive and requires a craniotomy. Recently, transgenic mice expressing channelrhodopsin-2 in neurons (Thyl-ChR2-YFP) have been used to induce CSD by applying blue light stimulation through the intact skull [38]. While obtaining transgenic mice presents an experimental challenge, this new method does not require a craniotomy and can induce CSD in a minimally invasive manner. Evaluation methods for CSD induced by either method include measurements of the threshold stimulus intensity required to induce CSD, the frequency of CSD, and the speed of CSD propagation.

## 5. What Are the Cellular Level Consequences of the Emergence of CSD?

The principal characteristics of CSD at the cellular level include: changes in the concentrations of potassium, sodium, calcium, and chloride ions; fluctuations in the direct current (DC) potential; and increases in extracellular levels of potassium, glutamate, and adenosine triphosphate (ATP). Concurrently, intracellular levels of sodium, calcium, and water increase rapidly. The excitatory amino acid glutamate, released extracellularly, binds to N-methyl-D-aspartate (NMDA) glutamate receptors, further enhancing the influx of sodium and calcium into cells [31,46,47].

This series of ionic gradient changes generates a slow DC potential shift outside the cell [10,48] and leads to the activation of the glial cell Na^+^/K^+^-ATPase pump. The reuptake of extracellularly released potassium ions buffers against ionic disturbances [37]. Moreover, glial cells absorb glutamate to avoid excitotoxicity [49].

The influx of water into cells due to CSD increases the volume of neurons, which in turn causes dendrites to retract and form blisters (blebs). Although the influx of sodium and calcium into cells results in an increase in intracellular osmotic pressure, potentially causing cytotoxic edema, most dendritic morphological changes are reversible. Following CSD, the neuronal cell body rapidly recovers its previous volume [10]. In contrast to neurons, glial cells demonstrate minimal CSD impairments, retain their functionality, and support neuronal recovery [43]. The activation of glial cells is significant in mitigating cytotoxicity; they help regulate extracellular ion concentrations and provide metabolic support to neurons. This glial response enhances overall neuronal health and facilitates recovery after CSD events.

## 6. Relationship of CSD to the Trigeminal Nervous System and Calcitonin Gene-Related Peptide (CGRP)

CSD is postulated to activate the trigeminal nervous system by altering concentrations of potassium ions and nitric oxide, leading to increased neuronal excitability. This suggests that CSD may be involved not only in the migraine aura but also in the headache phase, with neurogenic inflammation in the dura mater and metabolic alterations in the cerebral cortex occurring simultaneously with trigeminal system activation [43].

While the role of CGRP in migraine pathogenesis is well-established, the precise association between CSD and CGRP remains unclear. In a rat CSD model, a significant increase in CGRP mRNA levels was observed 24 h post-induction in the cerebral hemisphere in which CSD was induced. This indicates that CSD-induced reactive oxygen species (ROS) production initiates an inflammatory cascade that upregulates CGRP expression [44]. Additionally, an increase in the size of trigeminal ganglion neurons synthesizing CGRP mRNA was noted at 48 and 72 h after CSD induction [49]. These findings suggest that CSD contributes to migraine pain by enhancing CGRP gene expression in regions such as the cerebral cortex and trigeminal ganglion neurons.

The mechanism by which cortical events like CSD affect peripheral sensory neurons remains an area of active research. One proposed pathway involves the transport of solutes, including CGRP, from the cortex to the trigeminal ganglion via cerebrospinal fluid (CSF). Normally, the arachnoid membrane limits the direct action of cortical extracellular solutes on the dura mater. However, solutes may reach the dura mater indirectly through drainage into the dural lymphatic system via bridging veins. CSF is hypothesized to flow peripherally along cranial nerve sheaths, but tight junctions within these sheaths restrict the movement of CSF solutes into nerve tissue [50,51,52,53,54].

In a mouse model of classical migraine, Rasmussen et al. demonstrated that CSF transports CGRP and other solutes released during CSD to the extracellular space of the trigeminal ganglion, triggering trigeminal activation [55]. Their research showed that CSF flows into the trigeminal ganglion, facilitating nonsynaptic communication between the brain and trigeminal neurons. Following CSD onset, approximately 11% of the CSF proteome showed altered expression levels, with an increase in proteins that directly activate trigeminal ganglion receptors. CSF collected from animals experiencing CSD activated trigeminal neurons in naïve mice, partly due to CSF-borne CGRP. This discovery reveals a critical communication pathway between the central and peripheral nervous systems, potentially explaining how CSD can lead to migraine headaches.

Understanding the mechanism by which cortical events affect peripheral sensory neurons is crucial to developing targeted migraine therapies. Elucidating this connection could lead to novel interventions that disrupt the pathway between CSD-induced cortical changes and trigeminal system activation. Further research is essential to fully comprehend this mechanism and its implications for migraine treatment.

## 7. CSD and Threshold

There is a strong correlation between CSD and migraines. However, the specific factors that alter brain sensitivity to CSD and trigger its onset remain unclear. While animal studies often use KCl drops to induce CSD, this does not fully replicate clinical conditions. It is postulated that the genetic factors (such as channelopathies related to ion channels), physiological factors, and environmental influences that predispose individuals to migraines affect the threshold for CSD.

The relationship between genetic factors and the CSD threshold is illustrated by studies on familial hemiplegic migraine type 1 (FHM1). In FHM1 mouse models with R192Q and S218L mutations in the CACNA1A gene, the mice exhibited a significantly increased frequency and speed of CSD propagation compared to wild-type mice. Notably, female mice with these mutations showed higher CSD frequency and propagation rates than male mice, a difference not observed in wild-type mice [56]. These findings suggest that genetic mutations associated with migraine can lower the threshold for CSD, potentially contributing to migraine susceptibility.

Physiological factors, such as obesity, also influence CSD thresholds. Individuals with obesity and recurrent migraines often experience more severe headaches with increased sensitivity to light and sound [41,57,58]. In Zucker fatty (ZF) rats, an animal model of obesity, inducing CSD resulted in a significantly higher frequency of CSD events compared to wild-type rats (Figure 3) [41]. Chronic hyperleptinemia due to obesity and elevated levels of inflammatory cytokines like interleukin-6 (IL-6) and tumor necrosis factor-alpha (TNF-α) are hypothesized to lower the CSD threshold, making the brain more susceptible to migraine triggers.

A hallmark of migraine is recurrent headache attacks, which are known to be triggered by changes in climatic variables such as atmospheric pressure, humidity, and temperature [59,60]. In rats exposed to a hot environment that elevated their body temperatures, the induction of CSD led to a significant increase in the frequency of CSD occurrences compared to normal temperature conditions (Figure 4) [45]. This suggests that high ambient temperatures and increased body temperature may lower the threshold for CSD, potentially explaining why some individuals experience migraines in hot weather.

Understanding the factors that lower the CSD threshold has important clinical implications. Identifying genetic mutations that affect CSD susceptibility could lead to personalized treatment strategies targeting specific ion channels or signaling pathways. Similarly, recognizing the impact of constitutional and environmental factors may help in developing preventive measures, such as weight management programs for obese patients or advising susceptible individuals to avoid extreme temperatures.

By investigating the factors that influence the CSD threshold, we may gain valuable insights into migraine pathogenesis and pave the way for more effective, individualized therapeutic approaches.

## 8. CSD and Migraine Treatment

If lowering the CSD threshold triggers migraines, then raising the CSD threshold could potentially serve as a migraine treatment strategy. In a study using a rat CSD model, daily administration of conventional migraine prophylactic medications—such as topiramate, valproic acid, propranolol, and amitriptyline—for several months resulted in an increase in the CSD threshold and a dose-dependent reduction in the frequency of CSD by 40–80% [3]. Although these medications were ineffective as acute treatments and exhibited minimal efficacy when administered for short periods, prolonged treatment duration led to a more pronounced suppression of CSD [3]. These pharmacological agents are hypothesized to exert inhibitory effects on cortical excitability by regulating the expression of CSD-related ion channels, neurotransmitter receptors, and transporter genes.

In addition to conventional prophylactic agents, anti-CGRP antibodies have been investigated for their effects on CSD. In a rat model, intravenous administration of fremanezumab 4 h before CSD induction resulted in a reduction in the rate of CSD propagation and recovery time after CSD [42]. However, similar results were observed with isotype control antibodies, raising questions about the specific efficacy of anti-CGRP antibodies in inhibiting CSD.

In a rat CSD model, intraperitoneal administration of a CGRP receptor antagonist (MK-8825) 30 min before CSD induction significantly suppressed CSD-induced pain behaviors such as freezing and grooming in a dose-dependent manner, even though CSD itself was not suppressed [39]. In animal studies, “freezing” (a lack of movement) and “grooming” (excessive cleaning behaviors) are considered indicators of pain and discomfort. The reduction of these behaviors suggests effective analgesia. Histopathological analysis demonstrated that CGRP receptor antagonists indirectly suppressed c-Fos induction in the caudal subnucleus of the trigeminal spinal tract (TNC) and thalamic reticular nucleus (TRN), whereas no such effect was observed in the amygdala. These findings suggest that CGRP receptor antagonists may suppress trigeminal pain induced by CSD without inhibiting the CSD itself. Therefore, targeting CGRP may alleviate migraine pain by acting on trigeminal pathways, even if it does not suppress CSD directly.

As a promising novel therapeutic modality, intranasal administration of insulin-like growth factor 1 (IGF-1) has been explored for suppressing CSD [40,61]. IGF-1 markedly increased the CSD threshold and reduced CSD frequency without causing significant adverse effects [40,61]. IGF-1 is a polypeptide involved in the neurogenesis of neurons and glial cells and has been linked to various neurological functions. In rat models, intranasal administration of IGF-1 significantly increased the CSD threshold and reduced CSD frequency without causing significant adverse effects. A single dose of IGF-1 inhibited CSD within an hour and continued to provide protection for at least seven days [40]. A two-week course of IGF-1, administered every third day, further decreased CSD susceptibility and showed no aberrant effects on glial activation, nasal mucosa, or serum markers of toxicity [40]. Moreover, IGF-1 reduced oxidative stress and nociceptive activation in the trigeminal system, which is crucial in migraine pathophysiology [61]. Given its efficacy and lack of significant adverse effects, intranasal IGF-1 represents an easy-to-implement and promising therapeutic modality to mitigate susceptibility to CSD and migraine headaches.

Furthermore, the efficacy of vagus nerve stimulation (VNS) in suppressing CSD has been documented. In a rat model, both invasive VNS (iVNS) and non-invasive VNS (nVNS) reduced the frequency of CSD, increased the threshold, and decreased the propagation speed [36]. In one study, iVNS markedly increased the electrical CSD threshold by more than two-fold and decreased the frequency of KCl-induced CSD by 22% when applied to the intact vagus nerve [62]. The efficacy of iVNS was not affected by distal vagotomy but was completely negated by proximal vagotomy. Additionally, a pharmacological blockade of the nucleus tractus solitarius (NTS)—the primary relay for vagal afferents—using lidocaine or a glutamate receptor antagonist prevented the suppression of CSD by nVNS. These findings indicate that VNS exerts an inhibitory effect on CSD through the central afferents of the NTS, which project to subcortical neuromodulatory centers providing serotonergic and noradrenergic innervation to the cortex.

Understanding the connection between the CSD threshold and migraine pain is crucial. Raising the CSD threshold reduces the likelihood of CSD occurrence, thereby potentially decreasing the frequency and severity of migraines. By preventing the initiation of CSD, therapeutic strategies can interrupt the cascade of events leading to migraine aura and headache. This approach underscores the importance of developing treatments that modify cortical excitability and susceptibility to CSD as a means of managing migraine.

## 9. CSD and the Role of Epigenetics

Recent research has highlighted the significant role of both genetic and non-genetic factors in migraine susceptibility [63]. Genome-wide association studies (GWAS) have identified multiple genetic variants associated with common migraine. A recent large-scale GWAS involving over 102,000 migraine cases and 771,000 controls identified 123 loci associated with migraine susceptibility, of which 86 were previously unknown. These loci include genes involved in neuronal and vascular functions, such as HMOX2, CACNA1A, MPPED2, SPINK2, FECH, CALCA/CALCB (which encode calcitonin gene-related peptide), and HTR1F (serotonin 1F receptor). Collectively, these findings provide strong evidence that neurovascular mechanisms underlie migraine pathophysiology [64] and have enhanced our understanding of the genetic underpinnings of migraine, although the specific causal genes and mechanisms remain to be fully elucidated.

In addition to genetic predispositions, epigenetic mechanisms—heritable changes in gene expression that do not involve alterations in the DNA sequence—play a crucial role in modulating migraine susceptibility [65,66]. Epigenetic factors, such as DNA methylation, histone modifications, and non-coding RNAs, can influence gene expression in response to environmental stimuli. DNA methylation, which involves the addition of a methyl group to the cytosine base in DNA, typically acts to repress gene transcription. Histone modifications, such as acetylation and methylation, can alter chromatin structure and regulate gene expression, while non-coding RNAs can modulate gene expression post-transcriptionally. Factors like sex hormones, lifestyle choices, and environmental changes may influence migraine susceptibility through these epigenetic modifications, potentially modulating the expression of genes associated with migraine risk.

Although recent studies have explored DNA methylation differences between migraine sufferers and controls, significant loci have not been identified, largely due to the limitations of using peripheral blood samples instead of brain tissue [67,68]. Rodent models of migraine provide valuable insights, as they allow for the examination of tissue-specific methylation patterns in the brain, which may reflect those in humans [69]. These models are particularly relevant for studying neurological diseases like migraines because they enable direct access to brain tissues affected by CSD.

Furthermore, studies have shown that chronic administration of certain preventive migraine drugs, such as valproate and topiramate, can suppress CSD in preclinical models [3,70]. This suggests that these drugs might work by altering DNA methylation and affecting migraine-related genes over time, thereby modifying cortical excitability and susceptibility to CSD.

Vila-Pueyo et al. investigated the short-term effects of CSD on genome-wide DNA methylation, with and without prophylactic treatments using topiramate and valproate, in an animal model [71]. They found that CSD significantly altered the DNA methylation profile of the cortex. Notably, when treated with topiramate or valproate, different patterns emerged. Topiramate reduced the number of differentially methylated regions (DMRs) by nearly 50%, suggesting a stabilizing effect on the genome’s methylation status. In contrast, valproate increased the number of DMRs by 17% compared to the untreated group following CSD induction, indicating a different mechanism of action.

Most of these DMRs were located within intragenic regions, affecting gene bodies rather than promoters. Functional analysis revealed distinct over-represented pathways: protein processing in the untreated CSD group, metabolic processes in the topiramate-treated group, and synapse-related functions along with ErbB signaling, mitogen-activated protein kinase (MAPK) pathways, and retrograde endocannabinoid signaling in the valproate-treated group [71]. The ErbB pathway is involved in cell growth and differentiation, the MAPK pathway plays a role in transmitting chemical signals from the cell surface to the DNA in the cell nucleus, and retrograde endocannabinoid signaling modulates neurotransmitter release [71]. These differences suggest that topiramate and valproate may exert their prophylactic effects through distinct epigenetic mechanisms, potentially influencing gene expression related to neuronal excitability and synaptic function.

Understanding the connection between CSD-induced DNA methylation changes and migraine is crucial. CSD may trigger epigenetic modifications that alter gene expression patterns, contributing to migraine susceptibility and chronicity [67]. By identifying specific epigenetic changes associated with migraine, new therapeutic targets may emerge, such as drugs that modify DNA methylation or histone acetylation [67].

While DNA methylation has been a primary focus, other epigenetic mechanisms, such as histone modifications and non-coding RNAs, are also being studied in migraine research [67]. Histone modifications can influence chromatin structure and gene expression, potentially affecting neuronal function and pain pathways.

In conclusion, epigenetic modifications, particularly DNA methylation changes induced by CSD, play a significant role in migraine pathophysiology [67,71]. Elucidating these mechanisms may lead to novel therapeutic strategies targeting epigenetic factors, offering hope for more effective and personalized migraine treatments in the future.

## 10. Conclusions

This review provides a comprehensive analysis of the molecular and cellular mechanisms underlying CSD and its critical role in migraine pathogenesis. By examining the intricate processes of ionic imbalances, excitotoxicity, and neuronal-glial interactions, and epigenetic modifications, we highlight how CSD contributes to the onset and propagation of migraine aura and pain through the activation of the trigeminal nervous system and upregulation of CGRP.

We have discussed how factors such as genetic predisposition, obesity, epigenetic factors, and environmental conditions influence the threshold for CSD initiation, presenting potential targets for therapeutic intervention. While current treatments—including prophylactic agents and CGRP-targeting drugs—have improved the quality of life for many patients, their limited efficacy in certain individuals underscores the need for novel therapeutic strategies.

Emerging therapies, such as intranasal insulin-like growth factor 1 administration, epigenetic modulators, and vagus nerve stimulation, offer promising avenues for reducing CSD susceptibility and migraine frequency. Future research should focus on these innovative approaches and further elucidate the molecular pathways involved in CSD to develop more effective treatments.

By integrating historical context with current scientific findings, we aim to deepen the understanding of migraine pathophysiology and stimulate ongoing research in this field. We hope that this review serves as a valuable resource for researchers and clinicians dedicated to advancing migraine management and improving patient outcomes.

## Figures and Tables

**Figure 1 ijms-25-11163-f001:**
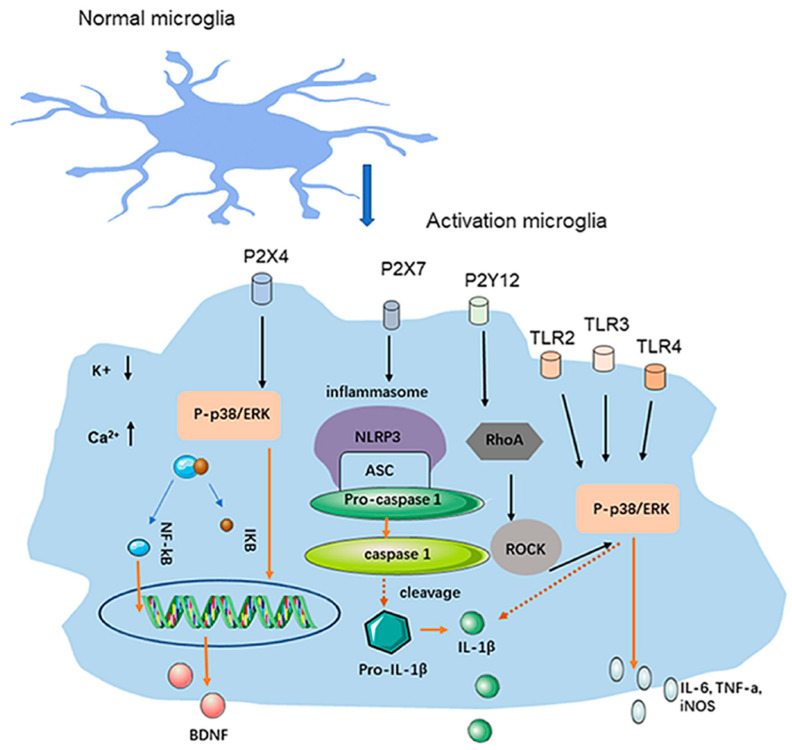
Microglial activation and proinflammatory cytokines. Microglia in the TNC release inflammatory factors and neuromodulators through receptors [32].

**Figure 2 ijms-25-11163-f002:**
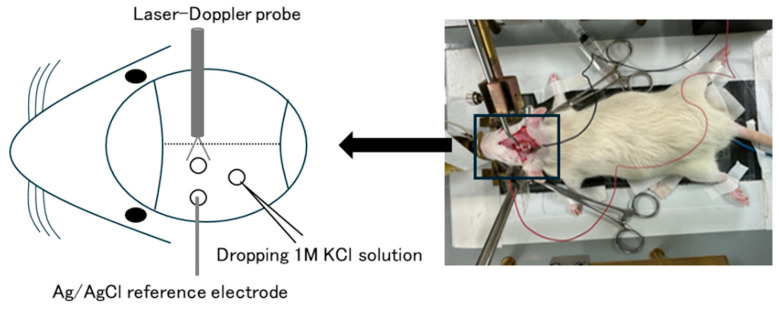
KCl-induced rat CSD model. In each rat, a tracheostomy is performed with controlled ventilation and under intraperitoneal anesthesia. After catheterization of the right femoral artery, the heart rate and mean arterial blood pressure are measured through a pressure transducer. A laser-Doppler probe is placed in the cerebral cortex through a left bone fenestration located 2 mm posterior and 2 mm left to the bregma to measure the CBF. A platinum electrode coated with chloroplatinic acid is used to measure the direct current (DC) potential, which is inserted through a left bone fenestration located 2 mm posterior and 4 mm lateral (left) to the bregma. The Ag/AgCl reference electrode is inserted into the temporal muscle. The potential between a platinum electrode and an Ag/AgCl electrode placed in the temporal muscle is recorded during the experiment. One bone fenestration is opened at a site 7 mm posterior and 3 mm lateral (left) to the bregma to apply the KCl solution. Thirty minutes to an hour of stable potential is allowed after insertion of the DC potential electrode, and then a 1.0 M KCl solution is applied through the bone fenestration to the cortical surface to induce CSD.

**Figure 3 ijms-25-11163-f003:**
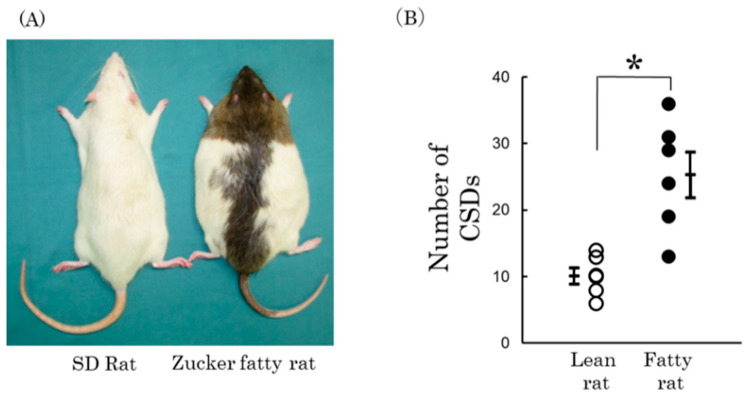
Changes in CSD due to constitutional factors (obesity). Zucker fatty rats (ZF rats) are homozygous for a mutated leptin receptor gene (fa gene) and show signs of obesity at 4 weeks of age, presenting a clear obese phenotype compared to normal rats (photo left: Sprague-Dawley rats; SD rats) (**A**). The number of CSDs in the ZF rat is significantly higher than in the Zucker lean rat, which does not have the gene mutation (**B**). * *p* < 0.01. Open circles represent lean rats, and closed circles represent fatty rats. Data from Kitamura et al. [41].

**Figure 4 ijms-25-11163-f004:**
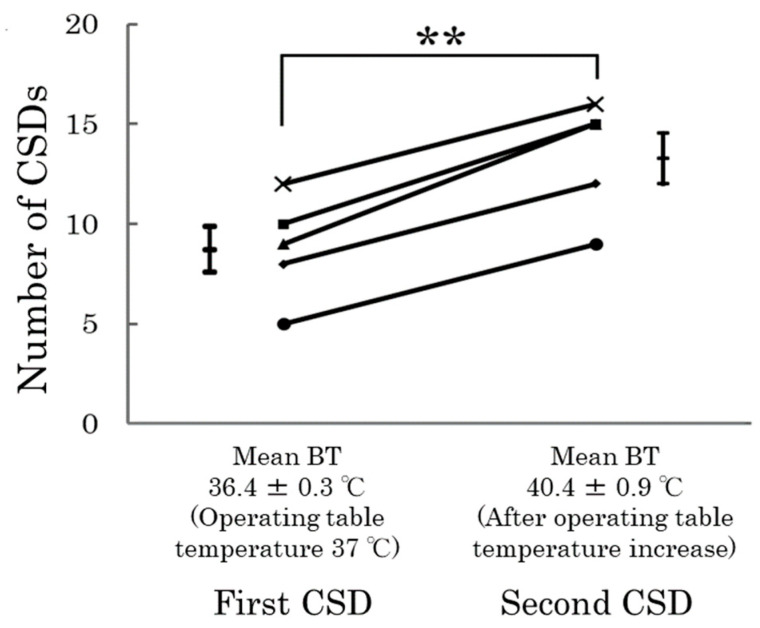
Changes in CSD due to environmental factors (temperature). When CSD is induced in rats whose body temperature (BT) has increased in a high-temperature (y) environment, the frequency of CSD increases significantly after the BT has increased (average BT, 40.4 °C) compared to before the BT increased (average BT, 36.4 °C). ** *p* < 0.001. Data from Kitamura et al. [45].

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
