# Peer review of "Molecular and Cellular Neurobiology of Spreading Depolarization/Depression and Migraine: A Narrative Review"

_ijms, 2024, doi:10.3390/ijms252011163_

Round 1

Reviewer 1 Report

Comments and Suggestions for Authors

Please find my comments and suggestions in the attached document.

Kind regards,

Review

Molecular and cellular neurobiology of spreading depolarization/depression and migraine: a narrative review

The text is well-structured and provides a comprehensive overview of the history and significance of cortical spreading depolarization/depression (CSD) and migraine. However, a few areas could be improved for clarity, readability, and completeness.

Section 2, lines 45-86:

1)     The text jumps from one historical point to another without smooth transitions. The reading flow of the manuscript could benefit from the addition of transitional sentences that may help the readers through the historical timeline and connect the various studies more harmoniously;

2)     It would be beneficial to introduce “MWoA” and “MWA” earlier in the text when discussing migraines with and without aura to maintain consistency;

3)     Some sentences are long and complex, making them difficult to follow. Breaking them into shorter sentences could improve readability.

Section 3, lines 88-120:

1)     The text is well-written and informative, explaining the role of microglia and neuroinflammation in the pathogenesis of CSD and migraines. If feasible, the manuscript could benefit from the inclusion of a figure/diagram or flowchart to represent the complex processes of microglial activation and neuroinflammation visually;

2)     Some points are repetitive without adding new information (lines 90-91 and 100-102) and could be combined to avoid redundancy;

3)     Review the text for minor grammatical issues, such as ensuring subject-verb agreement and the proper use of connectors.

Section 4, lines 123-158:

1)     The text mentions that CSD is not exclusive to migraines and can occur in other neurological conditions, but it does not fully elaborate on the significance of this finding. Providing more context about why this is important would enhance the text;

2)     Minor grammar issues are to be checked by the authors.

Section 5, lines 160-181:

1)     This section has some redundant sentences, particularly regarding the increase in extracellular potassium, glutamate, and ATP. Consider merging the duplicated information into one coherent statement;

2)     The text discusses the activation of glial cells and their role in mitigating cytotoxicity, but it could also approach the significance of these processes in the context of CSD. How does this impact the general outcome for neuronal health and recovery?

Section 6, lines 183-216:

1)     To clarify the CSF role, the authors could provide a concise explanation of how CSF carries solutes like CGRP to the trigeminal ganglion, underlining the importance of this mechanism in converging CSD to migraine;

2)     A brief discussion on the importance of understanding the mechanism by which cortical events affect peripheral sensory neurons and emphasizing the need for further research to elucidate this connection would strengthen the conclusion of this section;

3)     The text repeats some points, such as the role of CSD in migraine aura and headache phases. Review these issues to avoid redundancy.

Section 7, lines 219-264:

1)     Again, the text provides a well-supported discussion of the factors influencing the CSD threshold concerning migraines but repeats certain points. Review and consider combining similar points to make the text more concise and impactful;

2)     It is suggested that the authors discuss the potential implications of understanding CSD thresholds in clinical settings, such as personalized treatment strategies based on genetic or environmental factors.

Section 8, lines 265-313:

1)     Review the sentences from lines 272 to 276, as their content is repeated later in the same paragraph. Consider removing one of them to avoid redundancy. Similarly, the mention of IGF-1 having no significant adverse effects appears twice (lines 296 and 298); consider merging the sentences;

2)     In the discussion of CGRP receptor antagonists, consider briefly explaining why “freezing and grooming” behaviours are significant for understanding pain;

3)     It would be interesting to include in this section an explanation of the connection between the CSD threshold and migraine pain. It is suggested to explain why raising the threshold is thought to help reduce migraine severity;

4)     Ensure that abbreviations like “iVNS” (line 304) are defined when first mentioned;

5)     Review for minor grammatical errors, repetition of terms, extra spaces, question marks, etc., throughout the manuscript.

Section 9, lines 315-338:

1)     The introduction about genetic and non-genetic factors is good, but the authors could further elaborate on the key findings or progress in GWAS research, providing a more comprehensive understanding of the topic;

2)     The link between genetic variants and epigenetics could be made more substantial. After mentioning genetic predispositions, the authors could expand this topic regarding epigenetic mechanisms, such as DNA methylation, and how it may modulate gene expression and contribute to migraine susceptibility;

3)     The text refers to DNA methylation but does not clearly explain it. It is suggested to include a brief explanation in order to improve the technical accuracy of the manuscript;

4)     When the authors are discussing rodent models, in any section of the review, it is recommended to include why they are used; for example, the authors could mention that rodent models help because brain tissue samples can be used, which is relevant for studying neurological diseases like migraines, and so on;

5)     When discussing the effects of topiramate and valproate, the results are interesting but could be clearer. Explain why this difference matters or what it might suggest about the drugs’ mechanisms;

6)     The connection between CSD and DNA methylation could be more clearly explained. If possible, clarify why CSD-induced methylation changes are relevant for migraine research;

7)     Consider briefly explaining the significance of terms like “ErbB, MAPK, or retrograde endocannabinoid signaling pathways” (line 337);

8)     The section title is “CDS and the role of epigenetics,” but then the authors’ attention is given to DNA methylation alone. Consider to mention if other epigenetic mechanisms (e.g., histone modifications) are also being studied in migraine research;

9)     The authors are encouraged to add a concluding sentence synthesizing the key findings and their potential implications for future migraine treatments.

Comments on the Quality of English Language

The manuscript is well-written but could benefit from minor grammar and punctuation revisions.

Author Response

Response to Reviewer

We wish to express our gratitude to the reviewers for their kind comments. Please find our responses to these comments below.

Response to Reviewer 1

The text is well-structured and provides a comprehensive overview of the history and significance of cortical spreading depolarization/depression (CSD) and migraine. However, a few areas could be improved for clarity, readability, and completeness.

Section 2, lines 45-86:

Comment 1: The text jumps from one historical point to another without smooth transitions. The reading flow of the manuscript could benefit from the addition of transitional sentences that may help the readers through the historical timeline and connect the various studies more harmoniously.

Response: We appreciate the reviewer’s suggestion regarding the need for smoother transitions between historical points. We agree that the section would benefit from improved readability through better flow between the discussed studies. To address this, we have added transitional sentences that clearly guide the reader through the historical timeline of cortical spreading depolarization (CSD) and migraine research. These transitions highlight the connections between key findings while maintaining the chronological order of the discoveries.

Comment 2: It would be beneficial to introduce “MWoA” and “MWA” earlier in the text when discussing migraines with and without aura to maintain consistency.

Response: Thank you for this insightful suggestion. We have now introduced the abbreviations “MWoA” (migraine without aura) and “MWA” (migraine with aura) earlier in the manuscript to ensure consistency as followed: “Migraine can be divided into migraine with aura (MWA) and migraine without aura (MWoA).

Comment 3: Some sentences are long and complex, making them difficult to follow. Breaking them into shorter sentences could improve readability.

Response: We fully acknowledge the reviewer’s concern regarding the length and complexity of some sentences. We have carefully revised this section, breaking down longer sentences into shorter, more digestible ones. This revision improves the readability of the text while retaining the intended scientific meaning and nuance.

Section 3, lines 88-120:

Comment 1: The text is well-written and informative, explaining the role of microglia and neuroinflammation in the pathogenesis of CSD and migraines. If feasible, the manuscript could benefit from the inclusion of a figure/diagram or flowchart to represent the complex processes of microglial activation and neuroinflammation visually.

Response: Thank you for your positive feedback on the clarity and content of this section. We agree that a visual representation would enhance the understanding of the complex processes described. We have now included a figure that outlines the activation of microglia and its role in the neuroinflammatory cascade related to cortical spreading depolarization (CSD) and migraine pathogenesis (Figure 1).

Comment 2: Some points are repetitive without adding new information (lines 90-91 and 100-102) and could be combined to avoid redundancy.

Response: We appreciate the reviewer’s attention to redundancy. Upon review, we agree that the points in lines 90-91 and 100-102 overlap in content. To address this, we have combined the repetitive sections and streamlined the discussion to avoid unnecessary repetition, while ensuring that the key concepts are presented clearly and concisely.

Comment 3: Review the text for minor grammatical issues, such as ensuring subject-verb agreement and the proper use of connectors.

Response: Thank you for pointing out the grammatical issues. We have carefully reviewed the text and made the necessary corrections to ensure proper subject-verb agreement and the accurate use of connectors.

Section 4, lines 123-158:

Comment 1: The text mentions that CSD is not exclusive to migraines and can occur in other neurological conditions, but it does not fully elaborate on the significance of this finding. Providing more context about why this is important would enhance the text.

Response: We appreciate the reviewer’s observation and agree that further elaboration on the significance of CSD’s occurrence in other neurological conditions would enrich the manuscript. To address this, we have expanded the discussion to emphasize that the presence of CSD in conditions such as epilepsy, stroke, and traumatic brain injury suggests that it represents a more fundamental and generalized neurophysiological response: “The occurrence of CSD in diverse neurological conditions suggests that it may serve as a common pathway of pathological neuronal activity. In cerebral infarction, for example, CSD has been linked to the progression of ischemic damage by exacerbating neuronal depolarization and excitotoxicity. Similarly, in traumatic brain injury, CSD is associated with secondary injury processes that involve neuronal and glial dysfunction. This broader understanding of CSD underscores its potential as a therapeutic target, not only for migraines but also for conditions where excessive brain excitability or disruption of cellular homeostasis is a contributing factor. Investigating CSD in these various contexts may offer new avenues for therapeutic interventions aimed at stabilizing brain activity and preventing long-term damage across multiple neurological diseases.

Comment 2: Minor grammar issues are to be checked by the authors.

Response: Thank you for pointing out the minor grammatical issues. We have carefully reviewed this section for grammar and have corrected errors to ensure clarity and coherence throughout the text.

Section 5, lines 160-181:

Comment 1: This section has some redundant sentences, particularly regarding the increase in extracellular potassium, glutamate, and ATP. Consider merging the duplicated information into one coherent statement.

Response: Thank you for your feedback. I appreciate you pointing out the redundancies in the section. I have revised the text to eliminate duplication and improve clarity. Please find the updated version below: "The principal characteristics of CSD at the cellular level include changes in the concentrations of potassium, sodium, calcium, and chloride ions; fluctuations in the direct current (DC) potential; and increases in extracellular levels of potassium, glutamate, and adenosine triphosphate (ATP). Concurrently, intracellular levels of sodium, calcium, and water increase rapidly. The excitatory amino acid glutamate, released extracellularly, binds to N-methyl-D-aspartate (NMDA) glutamate receptors, further enhancing the influx of sodium and calcium into cells."

Comment 2: The text discusses the activation of glial cells and their role in mitigating cytotoxicity, but it could also approach the significance of these processes in the context of CSD. How does this impact the general outcome for neuronal health and recovery?

Response: Thank you for your insightful comment. You're absolutely right that the text could further elaborate on the significance of glial cell activation in the context of cortical spreading depression (CSD) and its impact on neuronal health and recovery. I've revised the text to address this point more thoroughly: “The activation of glial cells is significant in mitigating cytotoxicity; they help regulate extracellular ion concentrations and provide metabolic support to neurons. This glial response enhances overall neuronal health and facilitates recovery after CSD events.”

Section 6, lines 183-216:

Comment 1: To clarify the CSF role, the authors could provide a concise explanation of how CSF carries solutes like CGRP to the trigeminal ganglion, underlining the importance of this mechanism in converging CSD to migraine;

Response: Thank you for your valuable comments and suggestions. We added a concise explanation of how cerebrospinal fluid (CSF) carries solutes like CGRP to the trigeminal ganglion, emphasizing the importance of this mechanism in linking cortical spreading depression (CSD) to migraine: “The mechanism by which cortical events like CSD affect peripheral sensory neurons remains an area of active research. One proposed pathway involves the transport of solutes, including CGRP, from the cortex to the trigeminal ganglion via cerebrospinal fluid (CSF). Normally, the arachnoid membrane limits the direct action of cortical extracellular solutes on the dura mater; however, solutes may reach the dura mater indirectly through drainage into the dural lymphatic system via bridging veins. CSF is hypothesized to flow peripherally along cranial nerve sheaths, but tight junctions within these sheaths impede the unrestricted movement of CSF solutes into nerve tissue.”

Comment 2: A brief discussion on the importance of understanding the mechanism by which cortical events affect peripheral sensory neurons and emphasizing the need for further research to elucidate this connection would strengthen the conclusion of this section;

Response: We included a brief discussion on the importance of understanding how cortical events affect peripheral sensory neurons and emphasized the need for further research to elucidate this connection: “Understanding the mechanism by which cortical events affect peripheral sensory neurons is crucial for developing targeted migraine therapies. Elucidating this connection could lead to novel interventions that disrupt the pathway between CSD-induced cortical changes and trigeminal system activation. Further research is essential to fully comprehend this mechanism and its implications for migraine treatment.”

Comment 3: The text repeats some points, such as the role of CSD in migraine aura and headache phases. Review these issues to avoid redundancy.

Response: We reviewed the text to avoid repeating points about the role of CSD in migraine aura and headache phases, ensuring clarity and conciseness.

Section 7, lines 219-264:

Comment 1: Again, the text provides a well-supported discussion of the factors influencing the CSD threshold concerning migraines but repeats certain points. Review and consider combining similar points to make the text more concise and impactful;

Response: Thank you for your insightful comments. We reviewed the text to combine similar points and eliminate repetition, making the discussion more concise and impactful.

Comment 2: It is suggested that the authors discuss the potential implications of understanding CSD thresholds in clinical settings, such as personalized treatment strategies based on genetic or environmental factors.

Response: We added a discussion on the potential implications of understanding cortical spreading depression (CSD) thresholds in clinical settings, including personalized treatment strategies based on genetic and environmental factors: “Understanding the factors that lower the CSD threshold has important clinical implications. Identifying genetic mutations that affect CSD susceptibility could lead to personalized treatment strategies targeting specific ion channels or signaling pathways. Similarly, recognizing the impact of constitutional and environmental factors may help in developing preventive measures, such as weight management programs for obese patients or advising susceptible individuals to avoid extreme temperatures.

By investigating the factors that influence the CSD threshold, we may gain valuable insights into migraine pathogenesis and pave the way for more effective, individualized therapeutic approaches."

Section 8, lines 265-313:

Comment 1: Review the sentences from lines 272 to 276, as their content is repeated later in the same paragraph. Consider removing one of them to avoid redundancy. Similarly, the mention of IGF-1 having no significant adverse effects appears twice (lines 296 and 298); consider merging the sentences;

Response: We reviewed the sentences from lines 272 to 276 and merged to avoid repetition: “These pharmacological agents are hypothesized to exert inhibitory effects on cortical excitability by regulating the expression of CSD-related ion channels, neurotransmitter receptors, and transporter genes.” Similarly, we merged the sentences mentioning that IGF-1 has no significant adverse effects to avoid repetition” As a promising novel therapeutic modality, intranasal administration of insulin-like growth factor 1 (IGF-1) has been explored for suppressing CSD [40, 61]. IGF-1 markedly increased the CSD threshold and reduced CSD frequency without causing significant adverse effects [40, 61]. IGF-1 is a polypeptide involved in the neurogenesis of neurons and glial cells and has been linked to t various neurological functions. In rat models, intranasal administration of IGF-1 significantly increased the CSD threshold and reduced CSD frequency without causing significant adverse effects. A single dose of IGF-1 inhibited CSD within an hour and continued to provide protection for at least seven days [40]. A two-week course of IGF-1, administered every third day, further decreased CSD susceptibility and showed no aberrant effects on glial activation, nasal mucosa, or serum markers of toxicity [40]. Moreover, IGF-1 reduced oxidative stress and nociceptive activation in the trigeminal system, which is crucial in migraine pathophysiology [61]. Given its efficacy and lack of significant adverse effects, intranasal IGF-1 represents an easy-to-implement and promising therapeutic modality to mitigate susceptibility to CSD and migraine headaches.”

Comment 2: In the discussion of CGRP receptor antagonists, consider briefly explaining why “freezing and grooming” behaviours are significant for understanding pain;

Response: We added a brief explanation of why "freezing and grooming" behaviors are significant indicators of pain in animal studies, particularly in the context of CGRP receptor antagonists: “In animal studies, "freezing" (a lack of movement) and "grooming" (excessive cleaning behaviors) are considered indicators of pain and discomfort. The reduction of these behaviors suggests effective analgesia.”

Comment 3: It would be interesting to include in this section an explanation of the connection between the CSD threshold and migraine pain. It is suggested to explain why raising the threshold is thought to help reduce migraine severity;

Response: We included an explanation of why raising the CSD threshold is thought to help reduce migraine severity, highlighting its therapeutic implications:” Understanding the connection between the CSD threshold and migraine pain is crucial. Raising the CSD threshold reduces the likelihood of CSD occurrence, thereby potentially decreasing the frequency and severity of migraines. By preventing the initiation of CSD, therapeutic strategies can interrupt the cascade of events leading to migraine aura and headache. This approach underscores the importance of developing treatments that modify cortical excitability and susceptibility to CSD as a means of managing migraine.”

Comment 4: Ensure that abbreviations like “iVNS” (line 304) are defined when first mentioned;

Response: We ensured that abbreviations like "iVNS" (invasive vagus nerve stimulation) are defined upon their first occurrence: “In a rat model, both invasive VNS (iVNS) and non-invasive VNS (nVNS) reduced the frequency of CSD, increased the threshold, and decreased the propagation speed”.

Comment 5: Review for minor grammatical errors, repetition of terms, extra spaces, question marks, etc., throughout the manuscript.                                            

Response: We thoroughly reviewed the manuscript for minor grammatical errors, repetition of terms, extra spaces, and other typographical errors.

Section 9, lines 315-338:

Comment 1: The introduction about genetic and non-genetic factors is good, but the authors could further elaborate on the key findings or progress in GWAS research, providing a more comprehensive understanding of the topic;

Response: We expanded the discussion on the key findings and progress in genome-wide association studies (GWAS) related to migraine, providing a more comprehensive understanding of the genetic variants associated with the condition. “A recent large-scale GWAS involving over 102,000 migraine cases and 771,000 controls identified 123 loci associated with migraine susceptibility, of which 86 were previously unknown. These loci include genes involved in neuronal and vascular functions, such as HMOX2, CACNA1A, MPPED2, SPINK2, FECH, CALCA/CALCB (which encode calcitonin gene-related peptide), and HTR1F (serotonin 1F receptor). These findings provide strong evidence that neurovascular mechanisms underlie migraine pathophysiology.”

Comment 2: The link between genetic variants and epigenetics could be made more substantial. After mentioning genetic predispositions, the authors could expand this topic regarding epigenetic mechanisms, such as DNA methylation, and how it may modulate gene expression and contribute to migraine susceptibility;

Response: We enhanced the discussion on how epigenetic mechanisms, particularly DNA methylation, modulate gene expression and contribute to migraine susceptibility, bridging the gap between genetic predispositions and epigenetic influences.” In addition to genetic predispositions, epigenetic mechanisms—heritable changes in gene expression that do not involve alterations in the DNA sequence—play a crucial role in modulating migraine susceptibility. Epigenetic factors, such as DNA methylation, histone modifications, and non-coding RNAs, can influence gene expression in response to environmental stimuli. DNA methylation, which involves the addition of a methyl group to the cytosine base in DNA, typically acts to repress gene transcription. Factors like sex hormones, lifestyle choices, and environmental changes may influence migraine susceptibility through these epigenetic modifications, potentially modulating the expression of genes associated with migraine risk.”

Comment 3: The text refers to DNA methylation but does not clearly explain it. It is suggested to include a brief explanation in order to improve the technical accuracy of the manuscript;

Response: We added a concise explanation of DNA methylation to improve the technical accuracy and clarity of the manuscript: “DNA methylation, which involves the addition of a methyl group to the cytosine base in DNA, typically acts to repress gene transcription.”

Comment 4: When the authors are discussing rodent models, in any section of the review, it is recommended to include why they are used; for example, the authors could mention that rodent models help because brain tissue samples can be used, which is relevant for studying neurological diseases like migraines, and so on;

Response: We included a rationale for using rodent models in migraine research, highlighting their relevance in studying neurological diseases and the ability to access brain tissue samples:” These models are particularly relevant for studying neurological diseases like migraines because they enable direct access to brain tissues affected by CSD.”

Comment 5: When discussing the effects of topiramate and valproate, the results are interesting but could be clearer. Explain why this difference matters or what it might suggest about the drugs’ mechanisms;

Response: We elaborated on why the differences in DNA methylation caused by topiramate and valproate are significant, discussing what these findings might suggest about the drugs’ mechanisms of action: “thereby modifying cortical excitability and susceptibility to CSD. Notably, when treated with topiramate or valproate, different patterns emerged. Topiramate reduced the number of differentially methylated regions (DMRs) by nearly 50%, suggesting a stabilizing effect on the genome's methylation status. In contrast, valproate increased the number of DMRs by 17% compared to the untreated group following CSD induction, indicating a different mechanism of action.”; “Notably, when treated with topiramate or valproate, different patterns emerged. Topiramate reduced the number of differentially methylated regionsby nearly 50%, suggesting a stabilizing effect on the genome's methylation status. In contrast, valproate increased the number of DMRs by 17% compared to the untreated group following CSD induction, indicating a different mechanism of action.”

Comment 6: The connection between CSD and DNA methylation could be more clearly explained. If possible, clarify why CSD-induced methylation changes are relevant for migraine research;

Response: We explained why CSD-induced methylation changes are relevant for migraine research, emphasizing their potential role in migraine susceptibility and treatment:” Understanding the connection between CSD-induced DNA methylation changes and migraine is crucial. CSD may trigger epigenetic modifications that alter gene expression patterns, contributing to migraine susceptibility and chronicity. By identifying specific epigenetic changes associated with migraine, new therapeutic targets may emerge, such as drugs that modify DNA methylation or histone acetylation.”

Comment 7: Consider briefly explaining the significance of terms like “ErbB, MAPK, or retrograde endocannabinoid signaling pathways” (line 337);

Response: We briefly explained the significance of terms like ErbB, MAPK, and retrograde endocannabinoid signaling pathways: “Functional analysis revealed distinct over-represented pathways: protein processing in the untreated CSD group, metabolic processes in the topiramate-treated group, and synapse-related functions along with ErbB signaling, mitogen-activated protein kinase (MAPK) pathways, and retrograde endocannabinoid signaling in the valproate-treated group. The ErbB pathway is involved in cell growth and differentiation, the MAPK pathway plays a role in transmitting chemical signals from the cell surface to the DNA in the cell nucleus, and retrograde endocannabinoid signaling modulates neurotransmitter release.”

Comment 8: The section title is “CDS and the role of epigenetics,” but then the authors’ attention is given to DNA methylation alone. Consider to mention if other epigenetic mechanisms (e.g., histone modifications) are also being studied in migraine research;

Response: We noted that other epigenetic mechanisms, such as histone modifications, are also being studied in migraine research: “While DNA methylation has been a primary focus, other epigenetic mechanisms, such as histone modifications and non-coding RNAs, are also being studied in migraine research. Histone modifications can influence chromatin structure and gene expression, potentially affecting neuronal function and pain pathways.”

Comment 9:

The authors are encouraged to add a concluding sentence synthesizing the key findings and their potential implications for future migraine treatments.

Response: We included a concluding sentence synthesizing the key findings and their potential implications for future migraine treatments: “In conclusion, epigenetic modifications, particularly DNA methylation changes induced by CSD, play a significant role in migraine pathophysiology. Elucidating these mechanisms may lead to novel therapeutic strategies targeting epigenetic factors, offering hope for more effective and personalized migraine treatments in the future.”

Reviewer 2 Report

Comments and Suggestions for Authors

Molecular and Cellular Neurobiology of Spreading 2 Depolarization/Depression and Migraine: A Narrative Review

Summary: This is a review article on to Molecular and Cellular Neurobiology of Spreading 2 Depolarization/Depression and Migraine. However, the overall presentation, length and knowledge that it may provide is not significant. Therefore, I would not recommend it for possible publication in the IJMS. Instead, it may be published in some report or short paper in low impact journals. From the abstract no one can conclude if it is literature review or is it a report of some experimental study. From figure 1 it seems like it is a report of some experimental study. Finally, the conclusion states “This review outlines the history of CSD discovery and migraine pathogenesis with a focus on CSD” , which confirms it is a review of history (literature).

Major Issue:

1.      First, it is not clear if it is literature review or some comparative study because there are no statements how the research has been done. Looking to figure 1, it seems that there some study done by the authors (a primary study) and they want to report the findings. But this has not been described in abstract or introduction or any other place of the paper.

In case if it is literature review, which method did you follow for article searching, for selection of the articles, and data extractions from it? Please refer to SLR mythology for writing a review article.

2.      The introduction section has few statements as background and then only one statement to introduce the work done in this paper. i.e. “This review outlines the pathophysiology of migraine based on CSD, the relationship between CSD and 42 thresholds, and therapeutic interventions for CSD.” ___ I would like to see background and research significance, then proposed work along with method, and summary of findings and finally a list of key contributions.

3.      The paper is too short. A review article should have very strong information based on the existing work with some good categorization, a standard method of review, and different collected information in tabular and charted forms.

Minor Issues:

4.      The reference listed has been added twice

5.      It is not clear if figure 1 has been taken from other papers or is it made by authors during the current study (not previously published).

6.      Page 5, line 209: Reference is missing: Rasmussen et al. 208 demonstrated that the CSF (?)

7.      Page 4 line 179: As discussed in Chapter 3 (?): what does it indicate? Has this taken from some thesis where there are different chapters? Or what?

Comments on the Quality of English Language

Molecular and Cellular Neurobiology of Spreading 2 Depolarization/Depression and Migraine: A Narrative Review

Summary: This is a review article on to Molecular and Cellular Neurobiology of Spreading 2 Depolarization/Depression and Migraine. However, the overall presentation, length and knowledge that it may provide is not significant. Therefore, I would not recommend it for possible publication in the IJMS. Instead, it may be published in some report or short paper in low impact journals. From the abstract no one can conclude if it is literature review or is it a report of some experimental study. From figure 1 it seems like it is a report of some experimental study. Finally, the conclusion states “This review outlines the history of CSD discovery and migraine pathogenesis with a focus on CSD” , which confirms it is a review of history (literature).

Major Issue:

1.      First, it is not clear if it is literature review or some comparative study because there are no statements how the research has been done. Looking to figure 1, it seems that there some study done by the authors (a primary study) and they want to report the findings. But this has not been described in abstract or introduction or any other place of the paper.

In case if it is literature review, which method did you follow for article searching, for selection of the articles, and data extractions from it? Please refer to SLR mythology for writing a review article.

2.      The introduction section has few statements as background and then only one statement to introduce the work done in this paper. i.e. “This review outlines the pathophysiology of migraine based on CSD, the relationship between CSD and 42 thresholds, and therapeutic interventions for CSD.” ___ I would like to see background and research significance, then proposed work along with method, and summary of findings and finally a list of key contributions.

3.      The paper is too short. A review article should have very strong information based on the existing work with some good categorization, a standard method of review, and different collected information in tabular and charted forms.

Minor Issues:

4.      The reference listed has been added twice

5.      It is not clear if figure 1 has been taken from other papers or is it made by authors during the current study (not previously published).

6.      Page 5, line 209: Reference is missing: Rasmussen et al. 208 demonstrated that the CSF (?)

7.      Page 4 line 179: As discussed in Chapter 3 (?): what does it indicate? Has this taken from some thesis where there are different chapters? Or what?

Author Response

Response to Reviewer

We wish to express our gratitude to the reviewers for their kind comments. Please find our responses to these comments below.

Response to Reviewer 2

General comments: This is a review article on to Molecular and Cellular Neurobiology of Spreading 2 Depolarization/Depression and Migraine. However, the overall presentation, length and knowledge that it may provide is not significant. Therefore, I would not recommend it for possible publication in the IJMS. Instead, it may be published in some report or short paper in low impact journals. From the abstract no one can conclude if it is literature review or is it a report of some experimental study. From figure 1 it seems like it is a report of some experimental study. Finally, the conclusion states “This review outlines the history of CSD discovery and migraine pathogenesis with a focus on CSD” , which confirms it is a review of history (literature).

Response: Thank you very much for taking the time to review our manuscript and for providing your valuable feedback. We sincerely appreciate your insights, which are crucial for enhancing the quality and clarity of our work.

We acknowledge that the abstract and conclusion may not have sufficiently conveyed the nature and significance of our manuscript as a literature review. We apologize for any confusion this may have caused. To address this, we have revised both sections to clearly state the purpose and scope of the review, emphasizing its contribution to the field. The updated abstract is as follows:” Migraine is a prevalent neurological disorder, particularly among individuals aged 20–50 years, with significant social and economic impact. Despite its high prevalence, the pathogenesis of migraine remains unclear. In this review, we provide a comprehensive overview of cortical spreading depolarization/depression (CSD) and its close association with migraine aura, focusing on its role in understanding migraine pathogenesis and therapeutic interventions. We discuss historical studies that have demonstrated the role of CSD in the visual phenomena of migraine aura, along with modern imaging techniques confirming its propagation across the occipital cortex. Animal studies are examined to indicate that CSD is not exclusive to migraines but also occurs in other neurological conditions. At the cellular level, we review how CSD is characterized by ionic changes and excitotoxicity, leading to neuronal and glial responses. We explore how CSD activates the trigeminal nervous system and upregulates the expression of calcitonin gene-related peptides (CGRP), thereby contributing to migraine pain. Factors such as genetics, obesity, and environmental conditions that influence the CSD threshold are discussed, suggesting potential therapeutic targets. Current treatments for migraine, including prophylactic agents and CGRP-targeting drugs, are evaluated in the context of their expected effects on suppressing CSD activity. Additionally, we highlight emerging therapies such as intranasal insulin-like growth factor 1 and vagus nerve stimulation, which have shown promise in reducing CSD susceptibility and frequency. By elucidating the molecular and cellular mechanisms of CSD, this review aims to enhance the understanding of migraine pathogenesis and support the development of targeted therapeutic strategies:

he updated conclusion is as follows: “This review provides a comprehensive analysis of the molecular and cellular mechanisms underlying CSD and its critical role in migraine pathogenesis. By examining the intricate processes of ionic imbalances, excitotoxicity, and neuronal-glial interactions, and epigenetic modifications, we highlight how CSD contributes to the onset and propagation of migraine aura and pain through the activation of the trigeminal nervous system and upregulation of CGRP.

We have discussed how factors such as genetic predisposition, obesity, epigenetic factors, and environmental conditions influence the threshold for CSD initiation, presenting potential targets for therapeutic intervention. While current treatments—including prophylactic agents and CGRP-targeting drugs—have improved the quality of life for many patients, their limited efficacy in certain individuals underscores the need for novel therapeutic strategies.

Emerging therapies, such as intranasal insulin-like growth factor 1 administration, epigenetic modulators, and vagus nerve stimulation, offer promising avenues for reducing CSD susceptibility and migraine frequency. Future research should focus on these innovative approaches and further elucidate the molecular pathways involved in CSD to develop more effective treatments.

By integrating historical context with current scientific findings, we aim to deepen the understanding of migraine pathophysiology and stimulate ongoing research in this field. We hope that this review serves as a valuable resource for researchers and clinicians dedicated to advancing migraine management and improving patient outcomes.”

We would like to clarify that Figure 1 in our manuscript is derived from our previously published research. We have now ensured that this is clearly indicated in the manuscript with appropriate citations to prevent any confusion about the inclusion of experimental data in a review article.

Major Issue:

Comment 1: First, it is not clear if it is literature review or some comparative study because there are no statements how the research has been done. Looking to figure 1, it seems that there some study done by the authors (a primary study) and they want to report the findings. But this has not been described in abstract or introduction or any other place of the paper.

In case if it is literature review, which method did you follow for article searching, for selection of the articles, and data extractions from it? Please refer to SLR mythology for writing a review article.

Response: We apologize for any confusion. This manuscript is a narrative review summarizing the literature concerning cortical spreading depolarization (CSD) and its relation to migraines. The introduction have been revised to clearly state that this work is a review. Figure 1.in question represents an experimental method that has been established to stably induce cortical spreading depolarization (CSD) and was included without a proper reference number, which may have caused misunderstanding. We clarify that this method has already been published in an accepted paper (Figure 1 has been added, so the previous Figure 1 has been changed to Figure 2.): “Figure 2 shows a rat model of CSD induced by the application of KCl drops onto the cortical surface [41, 45].”

Comment 2: The introduction section has few statements as background and then only one statement to introduce the work done in this paper. i.e. “This review outlines the pathophysiology of migraine based on CSD, the relationship between CSD and 42 thresholds, and therapeutic interventions for CSD.” ___ I would like to see background and research significance, then proposed work along with method, and summary of findings and finally a list of key contributions.

Response: Thank you for your constructive feedback. We agree that the introduction would benefit from a more detailed background, a discussion of the research significance, an outline of our proposed work and methodology, a summary of findings, and a list of key contributions. We have revised the introduction accordingly to address these points: “Migraine is one of the most common neurological disorders, affecting over one billion individuals worldwide, with an estimated global prevalence of approximately 15%-20% [1-3]. It is particularly prevalent among ndividuals in their working age, typically between the aged 20–50 years and is the leading cause of disability among women under 50 years of age, significantly impacting their social and professional lives [1, 4]. Despite its high prevalence and substantial socioeconomic burden, the pathogenesis of migraine remains incompletely understood [5].

Migraine is a complex neurological condition characterized by recurrent episodes of severe headache, often accompanied by nausea, photophobia, and phonophobia. In approximately one-third of patients, these headaches are preceded by an aura phase lasting between 5 and 60 minutes [2]. The most common type of aura is visual, experienced by more than 90% of migraineurs with aura [2]. The aura has traditionally been closely associated with cortical spreading depolarization/depression (CSD), a wave of neuronal and glial depolarization that propagates across the cortex [3, 6-11]. CSD is not only implicated in the aura phase but is also thought to play a critical role in triggering migraine headaches by activating trigeminovascular pathways.

Understanding the role of CSD in migraine pathophysiology is crucial for developing targeted therapeutic interventions. While significant progress has been made in elucidating the mechanisms underlying CSD and its relationship with migraine, gaps remain in our knowledge of the factors that modulate the susceptibility to CSD and how this can be leveraged for treatment [6, 7]. Identifying these factors could lead to personalized medicine approaches, improving prophylactic and acute treatment strategies for migraine sufferers.

Aim of the Review

This narrative review aims to provide a comprehensive overview of the current understanding of migraine pathophysiology with a focus on the role of CSD. We explore the factors influencing the threshold for CSD initiation, including genetic predispositions, constitutional factors, and environmental triggers. Furthermore, we discuss therapeutic interventions that target CSD, examining both established and emerging treatments. By synthesizing current research on CSD and its impact on migraine, we aim to enhance the understanding of migraine pathophysiology and support the development of more effective, personalized therapeutic interventions.”

Comment 3: The paper is too short. A review article should have very strong information based on the existing work with some good categorization, a standard method of review, and different collected information in tabular and charted forms.

Response: We have expanded several sections, particularly those addressing the significance of CSD in neurological conditions beyond migraines (Section 4): “The occurrence of CSD in diverse neurological conditions suggests that it may serve as a common pathway of pathological neuronal activity. In cerebral infarction, for example, CSD has been linked to the progression of ischemic damage by exacerbating neuronal depolarization and excitotoxicity. Similarly, in traumatic brain injury, CSD is associated with secondary injury processes that involve neuronal and glial dysfunction. This broader understanding of CSD underscores its potential as a therapeutic target, not only for migraines but also for conditions where excessive brain excitability or disruption of cellular homeostasis is a contributing factor. Investigating CSD in these various contexts may offer new avenues for therapeutic interventions aimed at stabilizing brain activity and preventing long-term damage across multiple neurological diseases.” Additional details on personalized therapeutic strategies related to CSD thresholds (Section 7) have also been incorporated: “Understanding the factors that lower the CSD threshold has important clinical implications. Identifying genetic mutations that affect CSD susceptibility could lead to personalized treatment strategies targeting specific ion channels or signaling pathways. Similarly, recognizing the impact of constitutional and environmental factors may help in developing preventive measures, such as weight management programs for obese patients or advising susceptible individuals to avoid extreme temperatures.

By investigating the factors that influence the CSD threshold, we may gain valuable insights into migraine pathogenesis and pave the way for more effective, individualized therapeutic approaches.”

Minor Issues:

Comment 4: The reference listed has been added twice

Response: Thank you for catching this. We have carefully reviewed the references and corrected any duplicates.

Comment 5: It is not clear if figure 1 has been taken from other papers or is it made by authors during the current study (not previously published).

Response: We clarify that this method has already been published in an accepted paper (Figure 1 has been added, so the previous Figure 1 has been changed to Figure 2): “Figure 2 shows a rat model of CSD, as described by the authors, which is induced by the administration of KCl drops [41, 45]."”

Comment 6: Page 5, line 209: Reference is missing: Rasmussen et al. 208 demonstrated that the CSF (?)

Response: We sincerely apologize for any confusion caused by the inclusion of the phrase “CSF (?).” We added the reference number: “In a mouse model of classical migraine, Rasmussen et al. demonstrated that CSF transports CGRP and other solutes released during CSD to the extracellular space of the trigeminal ganglion, triggering trigeminal activation [55].”

Comment 7: Page 4 line 179: As discussed in Chapter 3 (?): what does it indicate? Has this taken from some thesis where there are different chapters? Or what?

Response: We sincerely apologize for any confusion caused by the inclusion of the phrase “Chapter 3 (?).” This was an oversight on our part, as it is a remnant from an earlier draft, and should have been removed during the revision process. We have now corrected this and removed the unnecessary reference to “Chapter 3 (?)” from the manuscript. Thank you for bringing this to our attention.

Response to Reviewer 3

This is an informative and well-covered paper on CSD and migraine. I have some comments:

General comments: This is a described as a narrative review. Anyhow, there seems to some formerly unpublished work in the reviewe, i.e. tha rat model in Figure 1. This is interesting and adds some value, but in some terms the "narrative" is misleading. There is also own work presented in Figure 3, but I understand this has been published. For the reader of a review, this is somewhat confusing. Please consider whether this should be in the paper.

Please define abbreviations in full text first time it is in the text.

Response: Thank you very much for your valuable feedback. We appreciate your insight regarding the inclusion of our own work in Figures 1 and 3 within the narrative review. We would like to clarify that Figure 1 is derived from our previously published research, which we have now properly cited in the manuscript to avoid any confusion. We understand that including unpublished data in a review can be misleading, and we apologize for any misunderstanding this may have caused.

Regarding the abbreviations, thank you for highlighting this issue. We have created an abbreviations list and have ensured that all abbreviations are defined in full upon their first mention in the text to enhance clarity for our readers.

Specific comments:

Comment 1: Line 33- economically active people might be misleading. I guesss the authors mean people in working ages? Please clarifye.

Response: We have revised this phrase to "individuals in their working age, typically between 20-50 years old" to remove ambiguity.

Comment 2: Line 34 - please define what a "leading disorder" in women means, at least add a reference.

Response: The phrase has been clarified to highlight migraines as one of the most prevalent neurological disorders affecting women under 50, particularly impacting their quality of life. A reference has been added to support this response: “. It is particularly prevalent among ndividuals in their working age, typically between aged 20–50 years and is the leading cause of disability among women under 50 years of age, significantly impacting their social and professional lives [1, 4].”

Comment 3: Line 49 - please define "temporal extent of flashes".

Response: We revised the sentence to explain the mianing "temporal extent of flashes": “This finding marked a pivotal moment in the history of CSD research, as it linked visual aura with brain activity”

Comment 4: The number 3 part of the paper could be written more clearly, now there are some repetitions, or it could be written a bit more concise.

Response: We appreciate the reviewer’s attention to redundancy. Upon review, we agree that the points in lines 90-91 and 100-102 overlap in content. To address this, we have combined the repetitive sections and streamlined the discussion to avoid unnecessary repetition, while ensuring that the key concepts are presented clearly and concisely.

Comment 5: Lines 126 and 127 need some references.

Response: We have added the necessary references to support the points in these lines: “CSD was originally discovered in the context of epilepsy, and subsequent studies have demonstrated that it can occur in various other neurological disorders, including cerebral infarction, subarachnoid hemorrhage, and traumatic brain injury [8, 33-35].”

Comment 6: Line 179 - there is a refeerence to Chapter 3 and then a question mark. I guess these parts of the paper could not be defined as chapters.

Response: We sincerely apologize for any confusion caused by the inclusion of the phrase “Chapter 3 (?).” This was an oversight on our part, as it is a remnant from an earlier draft, and should have been removed during the revision process. We have now corrected this and removed the unnecessary reference to “Chapter 3 (?)” from the manuscript. Thank you for bringing this to our attention.

Comment 7: Line 209 - there is a question mark behind CSF.

Response: We sincerely apologize for any confusion caused by the inclusion of the phrase “CSF (?).” We added the reference number: “In a mouse model of classical migraine, Rasmussen et al. demonstrated that CSF transports CGRP and other solutes released during CSD to the extracellular space of the trigeminal ganglion, triggering trigeminal activation [55].”

Comment 8: Line 235 - about obesity and migraine, please add reference/s.

Response: We have added the necessary references to support the points in these lines:” Individuals with obesity and recurrent migraines often experience more severe headaches with increased sensitivity to light and sound [41, 57, 58].”

Comment 9: Line 252 - as above, reference/s needed for weather and temperature.

Response: We appreciate the reviewer's comment. It is well known that migraine attacks can be triggered by changes in weather conditions. While there is literature suggesting that temperature changes may affect the cortical spreading depolarization (CSD) threshold, we acknowledge that no specific studies directly address the effects of weather changes on CSD. We have revised the sentence and added the references A hallmark of migraine is recurrent headache attacks, which are known to be triggered by changes in climatic variables such as atmospheric pressure, humidity, and temperature [59, 60]”

Comment 10: Line 280- 281 - the sentence is not clear to me regarding "similar to".

Response: We have revised the sentence to clear “similar to":  “However, similar results were observed with isotype control antibodies, raising questions about the specific efficacy of anti-CGRP antibodies in inhibiting CSD.”

Comment 11: The section about VNS needs to be clarified as for abbreviations - iVNS could maybe be referred to as both intravenous and invasive.

Response: We ensured that abbreviations like "iVNS" (invasive vagus nerve stimulation) are defined upon their first occurrence: “In a rat model, both invasive VNS (iVNS) and non-invasive VNS (nVNS) reduced the frequency of CSD, increased the threshold, and decreased the propagation speed”.

Comment 12: Line 315 - CDS instead pf CSD, needs to changed.

Response: We appreciate the reviewer's comment. We have revised the title as “9. CSD and the role of epigenetics”

Reviewer 3 Report

Comments and Suggestions for Authors

This is an informative and well-covered paper on CSD and migraine. I have some comments: 

General comments: This is a described as a narrative review. Anyhow, there seems to some formerly unpublished work in the reviewe, i.e. tha rat model in Figure 1. This is interesting and adds some value, but in some terms the "narrative" is misleading. There is also own work presented in Figure 3, but I understand this has been published. For the reader of a review, this is somewhat confusing. Please consider whether this should be in the paper. 

Please define abbreviations in full text first time it is in the text. 

Specific comments: 

Line 33- economically active people might be misleading. I guesss the authors mean people in working ages? Please clarifye. 

Line 34 - please define what a "leading disorder" in women means, at least add a reference. 

Line 49 - please define "temporal extent of flashes". 

The number 3 part of the paper could be written more clearly, now there are some repetitions, or it could be written a bit more concise. 

Lines 126 and 127 need some references. 

Line 179 - there is a refeerence to Chapter 3 and then a question mark. I guess these parts of the paper could not be defined as chapters. 

Line 209 - there is a question mark behind CSF. 

Line 235 - about obesity and migraine, please add reference/s. 

Line 252 - as above, reference/s needed for weather and temperature. 

Line 280- 281 - the sentence is not clear to me regarding "similar to". 

The section about VNS needs to be clarified as for abbreviations - iVNS could maybe be referred to as both intravenous and invasive. 

Line 315 - CDS instead pf CSD, needs to changed. 

Comments on the Quality of English Language

I have no other comments than those above and moderate editing needed. 

Author Response

Response to Reviewer

We wish to express our gratitude to the reviewers for their kind comments. Please find our responses to these comments below.

Response to Reviewer 3

This is an informative and well-covered paper on CSD and migraine. I have some comments:

General comments: This is a described as a narrative review. Anyhow, there seems to some formerly unpublished work in the reviewe, i.e. tha rat model in Figure 1. This is interesting and adds some value, but in some terms the "narrative" is misleading. There is also own work presented in Figure 3, but I understand this has been published. For the reader of a review, this is somewhat confusing. Please consider whether this should be in the paper.

Please define abbreviations in full text first time it is in the text.

Response: Thank you very much for your valuable feedback. We appreciate your insight regarding the inclusion of our own work in Figures 1 and 3 within the narrative review. We would like to clarify that Figure 1 is derived from our previously published research, which we have now properly cited in the manuscript to avoid any confusion. We understand that including unpublished data in a review can be misleading, and we apologize for any misunderstanding this may have caused.

Regarding the abbreviations, thank you for highlighting this issue. We have created an abbreviations list and have ensured that all abbreviations are defined in full upon their first mention in the text to enhance clarity for our readers.

Specific comments:

Comment 1: Line 33- economically active people might be misleading. I guesss the authors mean people in working ages? Please clarifye.

Response: We have revised this phrase to "individuals in their working age, typically between 20-50 years old" to remove ambiguity.

Comment 2: Line 34 - please define what a "leading disorder" in women means, at least add a reference.

Response: The phrase has been clarified to highlight migraines as one of the most prevalent neurological disorders affecting women under 50, particularly impacting their quality of life. A reference has been added to support this response: “. It is particularly prevalent among ndividuals in their working age, typically between aged 20–50 years and is the leading cause of disability among women under 50 years of age, significantly impacting their social and professional lives [1, 4].”

Comment 3: Line 49 - please define "temporal extent of flashes".

Response: We revised the sentence to explain the mianing "temporal extent of flashes": “This finding marked a pivotal moment in the history of CSD research, as it linked visual aura with brain activity”

Comment 4: The number 3 part of the paper could be written more clearly, now there are some repetitions, or it could be written a bit more concise.

Response: We appreciate the reviewer’s attention to redundancy. Upon review, we agree that the points in lines 90-91 and 100-102 overlap in content. To address this, we have combined the repetitive sections and streamlined the discussion to avoid unnecessary repetition, while ensuring that the key concepts are presented clearly and concisely.

Comment 5: Lines 126 and 127 need some references.

Response: We have added the necessary references to support the points in these lines: “CSD was originally discovered in the context of epilepsy, and subsequent studies have demonstrated that it can occur in various other neurological disorders, including cerebral infarction, subarachnoid hemorrhage, and traumatic brain injury [8, 33-35].”

Comment 6: Line 179 - there is a refeerence to Chapter 3 and then a question mark. I guess these parts of the paper could not be defined as chapters.

Response: We sincerely apologize for any confusion caused by the inclusion of the phrase “Chapter 3 (?).” This was an oversight on our part, as it is a remnant from an earlier draft, and should have been removed during the revision process. We have now corrected this and removed the unnecessary reference to “Chapter 3 (?)” from the manuscript. Thank you for bringing this to our attention.

Comment 7: Line 209 - there is a question mark behind CSF.

Response: We sincerely apologize for any confusion caused by the inclusion of the phrase “CSF (?).” We added the reference number: “In a mouse model of classical migraine, Rasmussen et al. demonstrated that CSF transports CGRP and other solutes released during CSD to the extracellular space of the trigeminal ganglion, triggering trigeminal activation [55].”

Comment 8: Line 235 - about obesity and migraine, please add reference/s.

Response: We have added the necessary references to support the points in these lines:” Individuals with obesity and recurrent migraines often experience more severe headaches with increased sensitivity to light and sound [41, 57, 58].”

Comment 9: Line 252 - as above, reference/s needed for weather and temperature.

Response: We appreciate the reviewer's comment. It is well known that migraine attacks can be triggered by changes in weather conditions. While there is literature suggesting that temperature changes may affect the cortical spreading depolarization (CSD) threshold, we acknowledge that no specific studies directly address the effects of weather changes on CSD. We have revised the sentence and added the references A hallmark of migraine is recurrent headache attacks, which are known to be triggered by changes in climatic variables such as atmospheric pressure, humidity, and temperature [59, 60]”

Comment 10: Line 280- 281 - the sentence is not clear to me regarding "similar to".

Response: We have revised the sentence to clear “similar to":  “However, similar results were observed with isotype control antibodies, raising questions about the specific efficacy of anti-CGRP antibodies in inhibiting CSD.”

Comment 11: The section about VNS needs to be clarified as for abbreviations - iVNS could maybe be referred to as both intravenous and invasive.

Response: We ensured that abbreviations like "iVNS" (invasive vagus nerve stimulation) are defined upon their first occurrence: “In a rat model, both invasive VNS (iVNS) and non-invasive VNS (nVNS) reduced the frequency of CSD, increased the threshold, and decreased the propagation speed”.

Comment 12: Line 315 - CDS instead pf CSD, needs to changed.

Response: We appreciate the reviewer's comment. We have revised the title as “9. CSD and the role of epigenetics”
